# Relationships between Spirituality, Religious Fundamentalism and Environmentalism: The Mediating Role of Right-Wing Authoritarianism

**DOI:** 10.3390/ijerph192013242

**Published:** 2022-10-14

**Authors:** Sebastian Binyamin Skalski, Teresa Loichen, Loren L. Toussaint, Patrycja Uram, Anna Kwiatkowska, Janusz Surzykiewicz

**Affiliations:** 1Faculty of Philosophy and Education, Catholic University of Eichstätt-Ingolstadt, 85072 Eichstätt, Germany; 2Faculty of Education, Cardinal Stefan Wyszyński University in Warsaw, 01938 Warsaw, Poland; 3Department of Psychology, Luther College, Decorah, IA 52101, USA; 4Institute of Psychology, Polish Academy of Sciences, 00950 Warsaw, Poland

**Keywords:** spirituality, religious fundamentalism, climate concern, pro-environmental behavior, right-wing authoritarianism

## Abstract

According to past research, religious attitudes can strongly influence individuals’ beliefs and behaviors. The purpose of this study was to assess the relationships between spirituality (the Scale of Spirituality; dimensions include religious spirituality, expanding consciousness, searching for meaning, sensitivity to art, doing good, and sensitivity to inner beauty), religious fundamentalism (the Religious Fundamentalism Scale), support for right-wing authoritarianism (the Right-Wing Authoritarianism Scale), climate concerns (the Environmental Concern Scale), and pro-environmental behavior (the Pro-Environmental Behavior Scale). The cross-sectional study involved 512 Poles aged 18–63 (M = 34.63, SD = 5.96; Mdn = 33), including 51% females. Multiple regression analysis revealed that two dimensions of spirituality (sensitivity to art and doing good) and religious fundamentalism are significant and opposite predictors of climate concern and pro-environmental behavior. Spirituality appeared to foster increased climate concern and caring behavior, while religious fundamentalism negatively predicted the same variables. Mediation analysis revealed that the relationship between religion and environmentalism could be explained in part by differences in support for right-wing authoritarianism (authoritarianism itself was negatively related to environmental outcomes). In addition, analysis of variance revealed that believers (70% of participants in the study were Catholic) showed significantly lower scores regarding climate concerns and pro-environmental behavior than non-believers, yet the inclusion of support for right-wing authoritarianism as a covariate in the equation reduced intergroup differences to statistical insignificance. The data obtained suggest that religious attitudes and socio-political views may play important roles in solving environmental problems.

## 1. Introduction

Religion is a source of both beliefs and moral values, and thus can have a strong influence on attitudes toward the environment and climate change [1,2,3]. As such, the study of religion and environmentalism should be an important component of the broader work of environmental humanities [4]. Within existing research, there are three broad and overlapping analytical approaches: (1) exegesis of scriptures and historical texts to uncover metaphysical and philosophical teachings about the environment, as well as clues to attitudes and behaviors in the past; (2) ethnographic studies of specific socio-religious and cultural practices related to the environment; and (3) socio-political analyses of religious movements and organizations that actively work to protect the environment [5]. Although much of this analysis points to the generally positive role that various religious beliefs can (or do) play in encouraging greater environmental sensitivity, some researchers and commentators formulate cautions and reservations. For example, Nelson [6] points out the limitations of textual analysis (as a method) in explaining real values and behavior. Alley [7] and Clements et al. [8] provide empirical data suggesting that some elements of religious beliefs and practices can be disrespectful and even harmful to the environment. Meanwhile, Tomalin [9] brings to the discussion the problem of whether religion or culture is at all an appropriate or sufficient framework through which to solve the enormous environmental problems faced by modern civilization.

The ongoing scientific discourse has so far failed to identify the exact place and role of religion in promoting pro-environmental behavior. There is a strong tradition in the social sciences, started with a classic essay by Lynn Jr. White, published in 1967, regarding the perceived negative role of the Judeo-Christian religion in creating pro-environmental attitudes. According to White, the current ecological crisis is a consequence of a religious worldview that assumes human dominance over nature, which allows people to exploit it ruthlessly according to their needs [3]. Indeed, a number of studies have shown that people who identify with the Christian religion manifest less concern for the environment than those who are non-religious (e.g., [10,11]). In addition, Eom [12] noted that religious people prefer the external attribution of responsibility for climate change and are less likely to take pro-environmental actions.

On the other hand, the assumption that religiosity is exclusively opposed to concern for the environment has been challenged by many researchers. In previous reports, frequent prayer [13] and exploratory religiosity [14] have been found to be associated with greater concern for the environment. Djupe and Hunt [15] noted that religious views can encourage the promotion of environmentalism through central educational and pro-social values. Meanwhile, other researchers point to both positive and negative influences of different types of religious beliefs on climate concern [14,16,17].

Undoubtedly, the various forms of religious involvement observed in western Christianity may be relevant in predicting the impact of religion on attitudes toward the environment. It should be noted, however, that Christianity is only one of the world religions with which people can identify. Other religions, e.g., Islam, Hinduism, Buddhism, and Shintoism, may offer their followers other conceptions of nature and human–environment relations, more or less favorable to the environment (e.g., [9,18,19]). Nevertheless, in this study, we focus mainly on the Christian religion, and more specifically on followers of Catholicism.

The above literature review thus points to the apparent paradox of religious environmentalism, in which religion can influence, both positively and negatively, pro-environmental behavior. Preston and Shin [20] attempted to explain this apparent contradiction through two separate and opposing influences: spirituality [21] and religious fundamentalism [22].

In this study, we define spirituality in terms of the search for universal truth and a form of activity that enables people to discover meaning and significance in the surrounding world [23,24]. It is seen as a dimension resulting from, among other things, the search for the meaning of life, and can be based on both the religious and non-religious spheres [25]. 

Religious fundamentalism, on the other hand, is associated with society-wide obedience to its infallible, sacred texts and advocating for the imposition of its dogmas (as a framework for determining what is allowed and what is forbidden) on society as a whole [26]. The specific form of belief that characterizes fundamentalists is also associated with opposition to secular legislation, treating members of their own movements as a select elite and selectively choosing those elements of tradition that should be given special emphasis and those aspects of modernity that are acceptable and exploitable [27]. While religious fundamentalism represents a dogmatic and authoritarian approach to belief and, as such, should negatively predict environmentalism and reinforce climate change denial, spirituality [21] should reflect a personal attitude toward the divine and thus can positively predict a pro-environmental orientation. Preston and Shin [20] also note that spirituality can encourage climate care through greater empathic compassion for others, including stronger moral associations with empathy [28] and universal pro-social concerns [29]. To the extent that people with high levels of spirituality show stronger compassion and moral concern for others, they may also have deeper concerns about environmental destruction and the serious consequences of climate change. Previous reports have focused exclusively on the impact of spirituality and religious fundamentalism on environmental attitudes [20]. Similar studies on pro-environmental behavior have not been conducted. Further analysis is needed to understand the relationship between religion and climate change. In addition, it would be interesting in future research to consider authoritarian traits, which, in view of previous findings, could potentially mediate the relationship between religion and environmentalism [30,31].

Religion, and especially religious fundamentalism, is shown to be in close association with right-wing authoritarianism, i.e., support for authoritarian and conventional values [32]. Thus, authoritarian traits may prove to be important (along with religion) co-variables in predicting pro-environmental orientation. Adorno [33] defined the authoritarian personality as a syndrome of nine categories of attitudes and beliefs (i.e., conventionalism, authoritarian submissiveness, authoritarian aggression, anti-intraception, superstition and stereotypes, power and toughness, destructiveness and cynicism, projection, exuberant sexuality). Altemeyer [34] challenged the concept of generalized authoritarianism and proposed a distinction between right-wing and left-wing authoritarianism. The first type involves reliance on socially legitimized authorities at a particular time and in a particular country (e.g., reliance on the dominant religion, government, or law). Meanwhile, left-wing authoritarianism involves support for authoritarians who are dedicated to destroying the state order. Initial research on the relationship between religion and authoritarianism suggested that religious people may be characterized by higher levels of authoritarian traits and greater racial prejudice compared to non-religious people [33]. However, later research has shown that the relationship between religion and authoritarian personalities is more complex and does not necessarily point to religiosity as a source of prejudice. Kahoe [35] noted that people with mature and reflective religiosity are not characterized by authoritarianism, in contrast to people with immature and instrumental religiosity (i.e., treating religion objectively, as a means for their own ends), in whom authoritarian traits are prominent. Finally, Altemeyer and Hunsberger [22] conducted an extensive survey of adherents of the world’s leading religions and showed that right-wing self-ritualization characterizes religious people with a strong fundamentalist orientation. Data from this study suggest that religious fundamentalists perceive their beliefs as the absolute truth, which does not allow them to accept other points of view. The consequence of such a phenomenon is obedience to authority, conventionalism, and a sense of superiority towards differently thinking people. At the same time, Altemeyer and Hunsberger [22] showed that people with an open religious style that emphasizes complexity and a desire to find answers to faith doubts prefer non-authoritarian behavior. Similarly, in a study by Wink et al. [36], spirituality was negatively associated with authoritarianism among Protestants and Catholics. On the other hand, Stanley and Wilson’s meta-analysis [37] based on 53 independent studies indicates that support for right-wing authoritarianism may be one of the key barriers to faith and action on environmental issues. Researchers have consistently shown that people with high rates of support for right-wing authoritarianism are less convinced that climate change is happening or that humans are contributing to the problem, are more accepting of the exploitation of natural resources, and do not believe that pro-environmental actions have any benefits or that we should protect nature [38,39,40]. In other studies, support for right-wing authoritarianism mediated the relationship between religiosity and prejudice, which the researchers claimed was due to “cognitively rigid ideologies” among those highly religious [31,41]. It seems, then, that support for right-wing authoritarianism may similarly mediate the negative relationship between religion and environmentalism, as these views are linked to a range of anti-environmental sentiments and at the same time may be reinforced by some forms of religiosity.

The objective of this study was to assess the links between religion and environmentalism. Based on the above literature review, we formulated the following hypotheses: (1) spirituality is a positive and religious fundamentalism is a negative predictor of climate concern and pro-environmental behavior; (2) support for right-wing authoritarianism is negatively related to climate concern and mediates the relationship between spirituality/religious fundamentalism and climate concern/pro-environmental behavior.

## 2. Materials and Methods

### 2.1. Participants and Procedure

The study was approved by the Ethics Committee of the Institute of Psychology of the Polish Academy of Sciences in Warsaw (No. 14/05/2021). It involved 512 Poles (51% of whom were women) aged 18–63 (M = 34.63, SD = 5.96; Mdn = 33). Of the participants, 70% declared Catholic religious affiliation, 6% Orthodox, 3% Protestant, 2% Muslim, and 1% were Jehovah’s Witnesses, while 18% declared no religious affiliation. Participation in the study did not involve any criteria and was preceded by informed consent. It was conducted in May 2022 via the Prolific survey panel (data were collected in Google Forms and then exported to an aggregate spreadsheet, excluding data to identify participants). The survey consisted of questionnaires to measure spirituality, religious fundamentalism, support for right-wing authoritarianism, climate concerns, and pro-environmental behavior. The average time to participate in the survey was 15 min. Participants were paid GBP 2.50.

### 2.2. Measures

The Scale of Spirituality by Skowronski and Bartoszewski [21] has been used to measure spirituality. This Polish scale consists of 36 statements arranged into six factors: religious spirituality (*α* = 0.96; all Cronbach’s alpha coefficients are for the current study data), which is related to staying connected to a spiritual guru; spirituality as expanding consciousness (*α* = 0.83), which describes trying to understand (learn about) oneself and others; spirituality as searching for meaning (*α* = 0.86), which refers to seeking answers to existential questions; spirituality as sensitivity to art (*α* = 0.85), which describes spiritual experiences while participating in cultural events; spirituality as doing good (*α* = 0.83), which refers to principles of social coexistence and caring for loved ones; and sensitivity to inner beauty (*α* = 0.85), which assesses the validity of moral life choices and admiration for nature. The participant expresses their attitude towards statements on a four-point Likert scale, where 1 = “I strongly disagree”, and 4 = “I strongly agree”. Sample statements include “I believe that I am taken care of by a Higher Power” and “I am sensitive to the injustice of other people”.

The Religious Fundamentalism Scale by Altemeyer and Hunsberger [22], in Polish standardization [42], was used to assess religious fundamentalism. The tool examines the beliefs and ways of categorizing the world used by religious fundamentalists of various denominations and does not address political ideology and partisan preferences. The unidimensional scale consists of 20 questions (α = 0.90). Individuals express their attitude towards each statement on a nine-point Likert scale (from 1 = “I strongly disagree” to 9 = “I strongly agree”). Sample statements include “There is a religion on this earth that teaches, without error, God’s truth” and “God will punish most severely those who abandon his true religion”.

The Right-Wing Authoritarianism Scale by Altemeyer [43], in a 12-item version in Polish [44], was used to measure support for right-wing authoritarianism. The univariate scale (*α* = 0.79) addresses three key components: conventionalism (adherence to established, known social norms), submissiveness (toward authority figures recognized in one’s own group), and aggression (toward individuals who do not adhere to group norms). Individuals rate their attitudes towards each statement on a seven-point scale (1 = “I strongly disagree” to 7 = “I strongly agree”). Sample statements include “It would be best if newspapers were censored so that people would not be able to get hold of destructive and disgusting material” and “There are many radical, immoral people trying to ruin things; the society ought to stop them”.

The Environmental Concern Scale [45] was used to assess climate concern. The scale consists of nine statements and includes three components: affective, cognitive, and conative. Participants express their attitude towards each statement on a five-point Likert scale (1 = “I strongly disagree” to 5 = “I strongly agree”). The scale’s translation was carried out by two independent translators, followed by a comparison and analysis by an environmental science terminology specialist. All the adaptation procedures were conducted based on WHO guidelines for cross-cultural research (Sousa and Rojjanasrirat, 2011). Exploratory factor analysis (EFA) revealed a single-factor scale (*α* = 0.81), and this factor accounted for 57.43% of the variance. Sample statements include “I am afraid when I think about environmental conditions for future generations” (affective), “The great majority of people do not act in an environmentally responsible way” (cognitive), and “It is still true that politicians do much too little to protect the environment” (conative).

The Pro-Environmental Behavior Scale [46] was used to measure pro-environmental behavior. The tool includes 16 statements that identify pro-environmental behavior related to recycling, shopping, energy and water conservation, as well as transportation and mobility. In the original version of the scale, the author asked people to respond yes/no to each statement according to engaging in specific pro-environmental behavior. In our version, we decided to use a five-point Likert scale, where 1 = “definitely no” and 5 = “definitely yes”, making it possible to assess the intensity of each pro-environmental behavior. The translation of the tool occurred in the same way as the adaptation of the Environmental Concern Scale. EFA indicated a unidimensional measure (*α* = 0.79), and this single factor explained 51.82% of the variance. Sample statements include “How often do you turn off the lights when you leave a room?” and “During the past year, how often have you car-pooled?”.

### 2.3. Statistical Analysis

Statistical data analysis was conducted in IBM SPSS Statistics 27. Normality of distribution was verified using the Kolgomorov–Smirnov test. Levene’s test was used to verify the homogeneity of variance. The obtained results allowed for the application of parametric tests. Pearson’s correlation analysis and multiple regression analysis were used to determine the relations between the variables. Hayes’ Process macro (model 4) was used to verify potential mediation effects. One-way analysis of variance (with contrast analysis) and Student’s t-test were used to evaluate differences. The effect size was assessed on the basis of *R*^2^, partial eta squared (*η*^2^), and Cohen’s *d*. The significance level was determined at *p <* 0.050.

## 3. Results

Concern for the environment and pro-environmental behavior were statistically significantly positively related to spirituality as expanding consciousness, spirituality as searching for meaning, spirituality as sensitivity to art, spirituality as doing good, and sensitivity to inner beauty; and negatively related to religious fundamentalism and right-wing authoritarianism. The correlation coefficients, their significance, and averages and standard deviations are shown in Table 1. 

In this study, age was found to favor spirituality as sensitivity to art (*r* = 0.24, *p* < 0.001), spirituality as doing good (*r* = 0.15, *p* < 0.001), and sensitivity to inner beauty (*r* = 0.20, *p* < 0.001). In addition, women showed greater concern for the environment (t = −5.36, *p* < 0.001, *d* = −0.47; M_men_ = 31.09, *SD* = 5.07; M_women_ = 33.44, *SD* = 4.85) and more frequent pro-environmental behavior (t = −2.06, *p* = 0.040, *d* = −0.18; M_men_ = 55.42, *SD* = 10.49; M_women_ = 57.23, *SD* = 9.40) than men.

Non-believers showed lower rates of religious spirituality (*F*_(1,510)_ = 165.05, *p* < 0.001, *η*^2^ = 0.24; M_believers_ = 28.40, SD = 10.98; M_non-believers_ = 17.59, SD = 6.96), religious fundamentalism (*F*_(1,510)_ = 285.82, *p* < 0.001, *η*^2^ = 0.36; M_believers_ = 75.00, SD = 25.45; M_non-believers_ = 41.10, SD = 17.95), right-wing authoritarianism (*F*_(1,510)_ = 81.01, *p* < 0.001, *η*^2^ = 0.14; M_believers_ = 40.95, SD = 10.19; M_non-believers_ = 32.91, SD = 9.79), and also higher rates of environment concern (*F*_(1,510)_ = 8.72, *p* = 0.003, *η*^2^ = 0.02; M_believers_ = 31.72, SD = 5.56; M_non-believers_ = 33.05, SD = 4.30) and pro-environmental behavior (*F*_(1,510)_ = 19.50, *p* < 0.001, *η*^2^ = 0.04; M_believers_ = 54.67, SD = 9.99; M_non-believers_ = 58.52, SD = 9.54) than believers (specific faith did not differentiate the results in a statistically significant way). Intergroup differences between non-believers and believers in terms of climate concern (*p* = 0.090) and pro-environmental behavior *(p* = 0.051) decreased to statistically insignificant levels after including authoritarianism (*B* = −0.07, *SE* = 0.02; *t* = −3.16, *p* = 0.002 for concern; *B* = −0.09, *SE* = 0.04; *t* = −2.04, *p* = 0.042 for behavior) as a covariate in the analyses. 

Multiple regression analysis showed that spirituality as sensitivity to art and spirituality as doing good were statistically significant positive predictors, while religious fundamentalism was a significant negative predictor of climate concern and pro-environmental behavior. The models were of a good fit to the data and were able to explain 13% of the variance in each case. For climate concern, spirituality as doing good was found to be the strongest predictor, while religious fundamentalism best predicted pro-environmental behavior. Values of regression coefficients are presented in Table 2. It should be noted that in the case of pro-environmental behavior, the presence of spirituality factors as covariates in the model increased the predictive value of religious fundamentalism from *β* = −0.10 (*p* = 0.032; see “c Path” in Table 3) to *β* = −0.22 (*p* < 0.001), indicating a classical suppression effect.

A bootstrap sampling analysis (5000) with a 95% confidence interval indicated that right-wing authoritarianism is a statistically significant partial mediator in the relationships between religious spirituality, spirituality as expanding consciousness, spirituality as sensitivity to art, spirituality as doing good, sensitivity to inner beauty, and religious fundamentalism (as X variables) vs. climate concern and pro-environmental behavior (as Y outcome variables). The significance of mediation effects and the values of regression coefficients are shown in Table 3. For religious spirituality (but only for climate concern) and religious fundamentalism, the inclusion of a mediator in the model increased the strength of the relationship between the variables, indicating a suppression effect. In the remaining (significant) models, the inclusion of a mediator reduced the strength of the association between spirituality dimensions and ecological variables.

## 4. Discussion

With more than 80% of the world’s population identifying themselves as religious [47], religion can influence attitudes and actions toward climate change. Previous reports, however, have pointed to the paradox of religious environmentalism, in which religion has promoted both positive and negative pro-environmental behavior [14,16,20,48]. In this study, we set out to explain this contradictory relationship through the separate and opposite influences of spirituality and religious fundamentalism. As expected (hypothesis 1), all dimensions of spirituality other than religious spirituality were positively related to climate concerns and pro-environmental behavior, while religious fundamentalism was negatively related to these variables. The regression analysis showed that among the various dimensions of spirituality, factors such as spirituality as sensitivity to art and spirituality as doing good may be particularly important in predicting climate care (in addition to the negative impact of religious fundamentalism). The regression models obtained for climate concern and pro-environmental behavior proved to be a good fit to the data and explained 13% of the variance in each case. It should be noted that while religious spirituality was not significantly related to environmentalism, simultaneous control of support for right-wing authoritarianism (i.e., an ideological orientation characterized by rigid thinking and resistance to new ideas) increased the strength of the association between religious spirituality and climate concern to statistically significant level, indicating the presence of a suppression phenomenon. Religious spirituality, according to Skowronski and Bartoszewski [21], is closely related to participation in religious practices and rituals (such as church ceremonies). The activities described can sometimes indicate an instrumental treatment of spirituality and religion, as evidenced by this variable’s strong links to religious fundamentalism and support for right-wing authoritarianism. Thus, it seems that the simultaneous measurement of support for right-wing authoritarianism introduces an external variance in measurement error for the results of the measure of religious spirituality (measurement artifact variance). This means that the spirituality variables suppress the irrelevant variance in religious fundamentalism, thereby allowing it to correlate (in its more purified form) more strongly with behavior.

Support for right-wing authoritarianism partially explained the associations of spirituality (all dimensions except spirituality as sensitivity) and religious fanaticism versus climate concern and pro-environmental behavior (hypothesis 2). The effect was that more spiritual people were less likely to indicate support for right-wing authoritarianism, which consequently reinforced concern for the environment. Conversely, those declaring higher rates of religious fundamentalism were more likely to display authoritarian traits and challenge climate change. In a similar study by Preston and Shin [20], political conservatism partially explained some of the negative relationship between general religiosity and environmental attitudes but did not explain the opposing effects of spirituality and religious fundamentalism. Therefore, it seems that support for right-wing authoritarianism (though often linked to conservatism) is a more important political variable for predicting environmentalism. Preston and Shin [20] also pointed out that pro-environmental attitudes can be further explained (in addition to political views) by moral values related to environmental issues, particularly concerns about harm and justice.

In our study, support for authoritarianism accounted for differences in climate concern and pro-environmental behavior between believers and non-believers. Including this variable as a covariate in the analyses reduced intergroup differences to statistically insignificant levels. Thus, it seems that the lower ecological orientation in believers is not due to religion per se, but to related differences in political views (i.e., greater support for authoritarianism). Whether differences in support for authoritarianism are the result of selection bias (people who support authoritarianism tend to affiliate with similar believers) or they are a product of the experience of belonging to a faith will need to be examined in future work. In addition, women showed more concern for the environment than men, which is in line with the consensus in the literature [49]. Eisler et al. [50] indicate that although men present higher environmental knowledge, women are more strongly mobilized for ecological thinking and behavior. In addition, age has fostered spirituality, which also supports previous findings [51]. In the literature, spirituality has even been identified as a developmental factor in the aging process [52].

We explored the issue of the relationship between religion and environmentalism through individual differences in spirituality and religious fundamentalism. It should be noted that alternative components of religion can also guide positive and negative environmental attitudes in similar ways. However, we expect that if other types of religious factors predict environmental attitudes, the differences will be similar to the effects of spirituality and religious fundamentalism. In previous research, Muñoz-García [14] found that climate concerns are predicted by differences in exploratory religiosity and biblical literalism. Similarly, Johnson [16] observed differences between mystical and authoritarian conceptions of God in environmental attitudes. These variables appear to map differences in spirituality and religious fundamentalism. For example, concepts of a mystical God share an experiential component and the transcendent emotions of spirituality, while literal thinking and concepts of a strict God are important aspects of religious fundamentalism. Finally, it should also be noted that the negative relationship between religion and climate concern is stronger within Christian denominations, which also tend to hold more fundamentalist views [53,54].

This study introduces new data on the relationship between religion, politics, and environmentalism. Before making broader generalizations, though, it is important to consider some limitations in the analyses conducted. First, the sample consisted primarily of Catholics (according to the Central Statistical Office—GUS—in Poland, approximately 85% of the population is Catholic; [55]). In such a situation, the effect of the influence of religion (not significant in our study) may be underestimated. We recommend further studies in other populations to generalize the conclusions beyond the specific context of Polish Catholicism. Second, this was a cross-sectional correlational study. Thus, the causes and effects of the phenomena cannot be determined. In future studies, it would be interesting to use data from longitudinal studies or experimental techniques (e.g., assessing the impact of a spirituality development intervention on measures of environmentalism).

In view of the dangers posed by the devastation of the environment and the consequences caused by the use of technologies hostile to the Earth, it becomes necessary to pay more attention to environmental education, ecological thinking, or ecological philosophy. In the case of educational issues of ecophilosophy, it should be noted that it is primarily about the philosophical and pedagogical basis of pro-ecological education in the family, school, social organizations, or religion [56]. In general, however, pro-environmental activists present a critical, albeit not fully justified, attitude to religion (especially Christianity), since traditional Christian piety itself, seeking contact with God, independent of the world, through prayer, deeds, and asceticism, necessarily also leads to respect for created entities [57]. Our article fits in well with the theme of ecophilosophy and allows us to expand previous findings on the opposing effects of spirituality and religious fundamentalism. While there is ample evidence in the literature of the havoc that religious fundamentalism can wreak (as with many secular fundamentalisms, such as nationalism, fascism, and Stalinism), one can only hope that, in these times of global socio-ecological threat, the voice coming from the Parliament of the World’s Religions in Salt Lake City in 2015 will have staying power: “The future we will embrace will be a new ecological civilization and a world of peace, justice and sustainable development in which the diversity of life flourishes. We will build this future as one family within a larger earthly community”.

## 5. Conclusions

This is the first work to point out the opposing effects of spirituality and religious fundamentalism in predicting climate concern and pro-environmental behavior. Spiritualism is found to favor environmentalism, while religious fundamentalism has a negative relationship with climate care. In addition, our study showed the critical role of support for right-wing authoritarianism, understood as an ideological orientation distinguished by resistance to new ideas and rigid thinking in envisioning environmentalism. The data obtained suggest that socio-political views are at the root of the basis of religion, so that they can guide attitudes toward nature.

## Figures and Tables

**Table 1 ijerph-19-13242-t001:** Means and correlations (N = 512).

	M (SD)	1.	2.	3.	4.	5.	6.	7.	8.	9.
1. Religious spirituality	23.69 (10.85)	-								
2. Spirituality as expanding consciousness	13.80 (1.97)	0.10 *	-							
3. Spirituality as searching for meaning	13.73 (3.69)	0.19 ***	0.46 ***	-						
4. Spirituality as sensitivity to art	10.81 (3.19)	0.08	0.35 ***	0.39 ***	-					
5. Spirituality as doing good	16.28 (2.64)	0.23 ***	0.50 ***	0.31 ***	0.37 ***	-				
6. Sensitivity to inner beauty	20.49 (2.70)	0.08	0.56 ***	0.36 ***	0.44 ***	0.51 ***	-			
7. Religious fundamentalism	60.24 (28.07)	0.52 ***	−0.17 ***	0.03	−0.08	−0.01	−0.16 ***	-		
8. left-wing authoritarianism	37.45 (10.77)	0.39 ***	−0.18 ***	−0.01	−0.21 ***	−0.10 *	−0.20 ***	0.54 ***	-	
9. Climate concern	32.30 (5.09)	0.03	0.17 ***	0.17 ***	0.26 ***	0.29 ***	0.28 ***	−0.14 **	−0.18 ***	-
10. Pro-environmental behavior	56.35 (9.98)	−0.04	0.23 ***	0.16 ***	0.26 ***	0.25 ***	0.19 ***	−0.10 *	−0.17 ***	0.47 ***

* *p* < 0.05; ** *p* < 0.01; *** *p* < 0.001.

**Table 2 ijerph-19-13242-t002:** Spirituality and religious fundamentalism as predictors of ecological outcomes (N = 512).

	Climate Concern ^a^			Pro-Environmental Behavior ^b^		
	B	SE	Β	B	SE	β
1. Religious spirituality	0.03	0.03	0.07	0.06	0.06	0.07
2. Spirituality as expanding consciousness	−0.21	0.15	−0.08	0.48	0.29	0.09
3. Spirituality as searching for meaning	0.09	0.07	0.07	0.13	0.13	0.05
4. Spirituality as sensitivity to art	0.24	0.08	0.15 **	0.57	0.15	0.18 ***
5. Spirituality as doing good	0.43	0.12	0.23 ***	0.79	0.24	0.21 ***
6. Sensitivity to inner beauty	0.16	0.12	0.09	−0.36	0.23	−0.10
7. Religious fundamentalism	−0.02	0.01	−0.14 *	−0.08	0.02	−0.22 ***

* *p* < 0.05; ** *p* < 0.01; *** *p* < 0.001; ^a^ *F*_(7,504)_ = 10.75, *p* = 0.001, *R*^2^ = 0.13; ^b^ *F*_(7,504)_ = 10.47, *p* = 0.001, *R*^2^ = 0.13.

**Table 3 ijerph-19-13242-t003:** The mediating role of right-wing authoritarianism in relationships between spirituality/religious fundamentalism and ecological outcomes (N = 512).

	a Path			b Path			c Path			c′ Path			Indirect Effect and B (SE)	95% CI LOWER UPPER
	B	SE	β	B	SE	β	B	SE	β	B	SE	β		
Outcome: climate concern														
RS → RA → EC	0.38	0.04	0.39 ***	−0.11	0.02	−0.22 ***	0.01	0.02	0.03	0.05	0.02	0.11 *	−0.085 (0.019)	−0.122; −0.050
SE → A → EC	−0.98	0.24	−0.18 ***	−0.07	0.04	−0.1 6 ***	0.45	0.11	0.17 ***	0.38	0.11	0.15 **	0.021 (0.009)	0.007; 0.042
SS → RA → EC	−0.01	0.13	−0.01	−0.09	0.02	−0.18 ***	0.23	0.06	0.17 ***	0.23	0.06	0.17 ***	0.001 (0.008)	−0.015; 0.017
SA → RA → EC	−0.72	0.15	−0.21 ***	−0.06	0.02	−0.13 **	0.42	0.07	0.26 ***	0.37	0.07	0.23 ***	0.028 (0.010)	0.005; 0.047
SD → RA→ EC	−0.42	0.18	−0.10 *	−0.07	0.02	−0.15 **	0.57	0.08	0.29 ***	0.53	0.08	0.28 ***	0.016 (0.008)	0.003; 0.032
SB → RA → EC	−0.81	0.17	−0.20 ***	−0.06	0.02	−0.13 **	0.53	0.08	0.28 ***	0.48	0.08	0.25 ***	0.027 (0.001)	0.009; 0.047
RF → RA → EC	0.21	0.01	0.54 ***	−0.08	0.03	−0.17 ***	−0.02	0.01	−0.14 **	−0.01	0.01	−0.01	−0.115 (0.042)	−0.198; −0.033
Outcome: pro-environmental														
RS → RA → EB	0.38	0.04	0.39 ***	−0.16	0.04	−0.18 ***	−0.04	0.04	−0.04	0.03	0.04	0.03	−0.066 (0.019)	−0.104; −0.031
SE → RA → EB	−0.98	0.24	−0.18 ***	−0.12	0.04	−0.13 **	1.17	0.22	0.23 ***	1.06	0.22	0.21 ***	0.023 (0.009)	0.007; 0.044
SS → RA → EB	−0.01	0.13	−0.01	−0.15	0.04	−0.16 ***	0.42	0.12	0.16 ***	0.42	0.11	0.16 ***	0.001 (0.008)	−0.014; 0.015
SA → RA → EB	−0.72	0.15	−0.21 ***	−0.11	0.04	−0.12 **	0.80	0.13	0.26 ***	0.72	0.14	0.23 ***	0.025 (0.010)	0.010; 0.051
SD → RA → EB	−0.42	0.18	−0.10 *	−0.13	0.04	−0.14 **	0.93	0.16	0.25 ***	0.87	0.16	0.23 ***	0.015 (0.008)	0.003; 0.031
SB → RA → EB	−0.81	0.17	−0.20 ***	−0.12	0.04	−0.13 **	0.70	0.16	0.19 ***	0.61	0.16	0.16 ***	0.027 (0.010)	0.009; 0.049
RF → RA → EB	0.21	0.01	0.54 ***	−0.15	0.05	−0.16 ***	−0.03	0.02	−0.10 *	−0.01	0.01	−0.19 ***	−0.087 (0.027)	−0.142; −0.034

* *p* < 0.05; ** *p* < 0.01; *** *p* < 0.001; RS = religious spirituality, SE = spirituality as expanding consciousness, SS = spirituality as searching for meaning, SA = spirituality as sensitivity to art, SD = spirituality as doing good, SB = sensitivity to inner beauty, RF = religious fundamentalism, RA = right-wing authoritarianism, EC = climate concern, EB = pro-environmental behavior, a path = effect of the independent variable on the mediator, b path = effect of the mediator on the dependent variable, c path = effect of the independent variable on the dependent variable, c′ path = direct effect of the independent variable on the dependent variable while controlling for the mediator. Effects are adjusted for religious affiliation.

## Data Availability

The data presented in this study are available on request from the corresponding author.

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
