# Peer review of "Relationships between Spirituality, Religious Fundamentalism and Environmentalism: The Mediating Role of Right-Wing Authoritarianism"

_ijerph, 2022, doi:10.3390/ijerph192013242_

Round 1
Reviewer 1 Report
Overall, the manuscript is well written, well organized, and well executed. I do not have any major recommendations for revision, except for one small matter. The authors are aware of the limitations of their study and note them in the closing portion of the manuscript. However, there is another possible limitation which the authors did not note. In particular, the study was conducted within a specific context (namely, Poland), and that fact may also be a potential limitation of the study. I am not certain, for example, whether American Catholics would necessarily respond in the same fashion as Polish Catholics. They might, but they might not. Hence, I think the authors should also note that having conducted the study within a specific cultural context may potentially limit the generalizability of the findings they obtain.
Author Response
We added „We recommend further studies in other populations to generalize the conclusions beyond the specific context of Polish Catholicism.” (p. 9).
Reviewer 2 Report
This is an important article and contribution. As the authors themselves note, its sample size is somewhat restricted, and further contributions to other religious communities are needed to fully round out the evidence presented. I genuinely hope they will do so or challenge others to do so.
Its scientific rigor (data collection and analysis) is particularly outstanding. Its discussion of the importance of such an undertaking is equally admirable. Its conclusion that fundamentalist thinking (in this case Polish Roman Catholicism) perhaps by its very nature, inhibits commitment to positive and activist concerns for climate change and the environment is equally noteworthy, as is the understanding that its own conflict with greater spirituality and the environment is precisely the point of tension is a message that needs to be shared.
Author Response
We thank you for your positive review report and are enthusiastic about conducting further research on the relation between religion and environmentalism.
Reviewer 3 Report
This is an excellent and essential paper. Congratulations to the authors. I only have some minor suggestions for the authors to consider before printing:
Sometimes you write "climate concern," and sometimes "environmental concern." I think it will be clearer when you always use the phrase climate concern (of course, please don't change the valid name of the Environmental Concern Scale).
Please provide sample items for all used scales in the study. Now You showed only in the first measure.
Would you like to indicate the mean and SD of age in the abstract?
I especially like your conclusions and practical implications. Well done!
Author Response
Following the reviewer's comment, we changed the variable's name to "climate concern", added sample items, and supplemented the mean and SD of age in the abstract.
Reviewer 3 marked “Extensive editing of English…” However, the other two reviewers marked “fine/minor.” The paper has been reviewed three times by an author with 20 years of scientific publishing expertise and we have spell- and grammar-checked the document an additional time. Only a couple of additional, very minor edits were identified and corrected.